# Pembrolizumab Induced Recall Dermatitis Occurring 5 Years After Radiotherapy

**DOI:** 10.3390/reports7040091

**Published:** 2024-11-04

**Authors:** Caroline J. Cushman, Fuad Abaleka, Andrew F. Ibrahim, Kiran Yalamanchili, Seshadri Thirumala, Donald Quick

**Affiliations:** 1Department of Medicine, Texas Tech University Health Science Center, Lubbock, TX 79430, USA; caroline.cushman@ttuhsc.edu (C.J.C.); andrew.ibrahim@ttuhsc.edu (A.F.I.); 2Department of Medicine, Covenant Health, Lubbock, TX 79410, USA; fabaleka@soundphysicians.com (F.A.); yalkik@covhs.org (K.Y.); 3Department of Pathology, Covenant Health, Lubbock, TX 79410, USA; sthirumala@ameripath.com

**Keywords:** radiation recall, radiation recall dermatitis, RRD, immune checkpoint inhibitor, pembrolizumab, immune-related adverse event (irAE)

## Abstract

**Background and Clinical Significance:** Radiation recall dermatitis (RRD) following immune checkpoint inhibitor (ICI) therapy has been infrequently reported. **Case Presentation:** We present a 47-year-old female patient who developed RRD of the breast following three doses of pembrolizumab administered as an adjuvant treatment post-nephrectomy for Stage III renal cell carcinoma (RCC). Notably, the affected breast had previously undergone external beam radiotherapy 247 weeks earlier for Stage IA invasive ductal carcinoma. She had received no prior chemotherapy at any point. RRD manifested as breast induration, erythema, and peau d’orange, and contraction of breast volume was noted following three cycles of pembrolizumab on week 17 (400 mg dose every 6 weeks). The dermatitis responded rapidly to systemic corticosteroids and no treatment interruption was needed. **Conclusions:** To date, this is the longest reported interval from completion of radiotherapy to RRD. A literature search underscores the variability in presentation and management of ICI-associated RRD.

## 1. Introduction and Clinical Significance

Radiation recall dermatitis (RRD) is an uncommon inflammatory reaction of the skin at a previously irradiated site precipitated by the administration of a systemic agent. This complication of radiotherapy is distinguished from the common immediate dermatological effects experienced while receiving radiation, or within a few days of completion [1]. A large systematic review of RRD involving 129 cases from any anatomical site reported that median time from radiotherapy to drug exposure was 8 weeks (range 2–132 weeks), time from drug exposure to RRD was 5 days (range 2–56 days), and time to significant improvement with intervention was 14 days (range 7–49 days). The majority (55%) of the analyzed cases involved patients with a history of breast cancer, and correspondingly, the breast/chest wall constituted the most sites of RRD (47%) [2]. While RRD associated with conventional chemotherapeutic agents has long been recognized, reports of RRD associated with targeted therapy and immunotherapy have been reported far less frequently, although this number is likely to increase given the increasing role of these agents in oncology [3,4,5].

Immune checkpoint inhibitors (ICIs) are a novel class of immunotherapy drugs utilized for treatment of a broad spectrum of cancers. These humanized monoclonal antibodies target inhibitory receptors (CTLA-4, PD-1, LAG-3, TIM-3) or ligands (PD-L1) expressed on T lymphocytes, antigen presenting cells, and tumor cells and elicit an anti-tumor response by stimulating the immune system. Nevertheless, the improved overall survival is complicated by the manifestation of immune-related adverse effects (irAEs) [6,7]. ICIs are commonly utilized in the treatment of metastatic renal cell carcinoma (RCC) for both clear cell and non-clear cell histology. In addition, pembrolizumab has shown benefits in the adjuvant setting for Stage III RCC with clear cell histology as well as Stage II clear cell histology with high-risk features such as Grade 4 tumors [8].

Radiation recall associated with ICI therapy is uncommon. Pneumonitis has been most frequently reported, while only a few cases have been reported at other sites such as myelitis and dermatitis [5,9,10,11]. Here, we describe a case of RRD in a patient receiving ICI as adjuvant therapy for newly diagnosed RCC nearly five years following radiotherapy for early-stage breast cancer. To our knowledge, this is the longest interval from completion of radiotherapy to the onset of RRD.

## 2. Case Presentation

A 42-year-old female was diagnosed with right breast invasive ductal carcinoma in November 2018. The tumor was estrogen receptor positive, progesterone receptor positive, HER2 overexpression negative, <1% p53 by IHC, and KI-67 of 25%. Oncotype analysis of the tumor revealed a 15% score, indicating an estimated benefit of chemotherapy at less than 1%. She underwent a lumpectomy in January 2019 with clear margins and final pathological stage IA (pT1bN0M0).

Radiation therapy was initiated in April 2019 and completed in May 2019, delivering 50 Gy in 25 fractions to the right breast over 43 elapsed days. An additional 10 Gy in 5 fractions over 6 elapsed days was applied to the left-right site. Cumulative radiation dose to the right breast was 60 Gy and cumulative fractions of 30 over 50 days. Radiation was given via Elekta Infinity using 10× photons and 12 MeV electrons. A three-dimensional conformational technique with medial/lateral tangential fields was utilized with MLC for beam shaping and field-in-field to improve dose homogeneity.

Tamoxifen was initiated shortly after the lumpectomy, and the patient then underwent total abdominal hysterectomy and bilateral salpingo-oophorectomy (TAH-BSO) in May 2019 after completions of radiation therapy. Post TAH-BSO, hormonal therapy was switched to anastrozole and subsequently to letrozole. The patient was placed back on tamoxifen in January 2020 and completed a total of five years of hormonal therapy in 2023.

In September 2023, the patient received a diagnosis of clear cell type RCC in the left kidney. The patient underwent a left radical nephrectomy. The tumor measured 4.2 cm with histologic grade 4, extending into the renal vein and into perinephric adipose tissue. Tumor necrosis was present (5% of tumor mass). Lymphovascular invasion was also evident. One regional (renal hilar) lymph node was positive for tumor involvement. There was no evidence of distant metastases seen on a CT scan of the chest, abdomen, pelvis, or on brain MRI. Disease was stage III (pT3aN1M0).

Eight weeks following surgery, the patient began pembrolizumab as an adjuvant therapy with a dose of 400 mg given every 6 weeks. She initially tolerated treatment well with no significant adverse reactions until week 17 (1 week prior to fourth cycle of treatment), when the patient reported changes in the right breast, including visible shrinking of breast size, generalized induration, erythema, peau d’orange texture, skin wrinkling, and mild desquamation (Figure 1A). Biopsy of the breast showed evidence of background chronic dermal fibrosis from prior radiotherapy and new onset perivascular inflammation consisting of a predominant CD5 positive T-lymphocyte infiltration. There was no evidence of active malignancy (Figure 2A–C). Clinicopathological features were consistent with RRD.

The patient was managed with systemic corticosteroid, starting with prednisone at 1 mg/kg/day for three weeks, tapered to 20 mg/day for three weeks, then 10 mg/day for one week, and 5 mg/day for one week before discontinuation, resulting in rapid symptom improvement. Breast retraction and peau d’orange changes had nearly resolved after one week of steroids. Pembrolizumab therapy was continued without delay, although subsequent cycles were given at 3-week intervals with doses of 200 mg. After two additional cycles of therapy, the patient did not show any evidence of recurrent radiation recall or other significant adverse effects of the checkpoint inhibitor (Figure 1B).

## 3. Discussion

We present a case of RRD involving the breast, characterized by visible shrinking of breast size, generalized induration, erythema, peau d’orange, skin wrinkling, and mild desquamation. The changes occurred 17 weeks from the initiation of pembrolizumab, and nearly five years after completion of radiation therapy. The patient’s contralateral breast showed no clinical changes. On review of previously reported RRD following ICI, we found nine other cases meeting the requisite of new onset symptoms or signs of dermatitis longer than seven days from completion of radiotherapy and longer than one day following the initiation of post-radiotherapy ICI therapy. Cases in which patients received ICIs during radiotherapy were excluded. Including the present patient, ten cases were felt to meet these criteria [5,11,12,13,14,15,16]. In this small set, the time from radiotherapy to ICI administration varied from 2 weeks to 247 weeks (present case) with a median of approximately 18 weeks. The time from the initiation of ICI to clinical RRD ranged from 3 days to 110 weeks (median 3.5 weeks). Including the present case, the associated ICIs had included nivolumab [12], pembrolizumab [13,15,16], and cemiplimab [14]. In two cases, additional immunotherapy was utilized with nivolumab (ipilimumab and lirilumab, a KIR/killer cell Ig-like receptor inhibitor) [5].

Two cases of RRD occurred in patients previously receiving irradiation to the breast. Billena et al. [12] reported RRD in a patient with oligometastatic breast cancer five weeks following radiotherapy and one week after initiating nivolumab. The patient had also received neo-adjuvant and adjuvant chemotherapy [12]. In the other case, a patient presented with synchronous Stage IVB endometrial cancer and Stage IA ER/PR positive, HER2 negative infiltrating ductal carcinoma. In this incident, pembrolizumab and concomitant lenvatinib were initiated approximately nine months following radiotherapy. RRD developed approximately six months later. However, dermatitis resolved following discontinuation of lenvatinib alone and the patient was continued on pembrolizumab, thus raising the question of the true contribution of the ICI in the etiology in this incident [16].

Additional cases have been reported following pembrolizumab therapy. Wang reported an onset of RRD on the third day of pembrolizumab in a patient with small cell lung cancer. Chest irradiation had been completed six months previously [13]. Sandhu noted RRD on the face and neck six days after initiating pembrolizumab and three months following radiation therapy for squamous cell carcinoma of the tongue [15].

The pathophysiological mechanisms underlying RRD are not fully understood. Teng et al. [15] examined the pattern of radiation recall pneumonitis (RRP) induced by PD-1 blockade [17]. The clinical presentation is different from common radiation pneumonitis (RP) or RRP induced by cytotoxic drugs. Interestingly, it also has been observed that patients who exhibit some form of radiation recall, particularly pneumonitis, seem to exhibit more durable responses to their malignancy [5]. Whether this initial observation holds up over time remains to be seen. While RRD might suggest a vigorous immune response potentially correlating with therapeutic efficacy, establishing it as a surrogate marker requires substantial validation through longitudinal studies to understand its predictive reliability. Continued investigation into pathophysiology may allow for better prediction and hence prevention of radiation recall reactions at any site. Factors such as RT dosimetric parameters, the role of tumor-infiltrating lymphocytes (TILs), and the impact of levels of PD-L1 expression may aid in our understanding of who might be at higher risk for certain reactions and hence provide optimal prevention and management of this subset of irAEs. Caccavale et al. [18] emphasize the importance of considering the immunocompromised cutaneous district concept, which could also be pivotal in understanding localized immune reactions such as RRD [18].

The clinical presentation of ICI-associated RRD has ranged from a mild rash with erythema to a more severe desquamating rash with skin necrosis [15]. Recent studies, such as a case reported by Cosio et al. [19], have illustrated additional dermatological complications during ICI therapy, highlighting pyodermitis in a patient treated with nivolumab for non-small cell lung cancer, further underscoring the complex dermatological landscape associated with ICI-associated RRD [19]. Our patient exhibited severe localized peau d’orange texture, desquamation, and discoloration in the irradiated area, along with breast shrinkage, indicative of advanced radiation recall dermatitis (RRD). A biopsy revealed significant infiltration of CD5^+^ T cells within the dermis, suggesting a heightened immune response, likely triggered by pembrolizumab. CD5^+^ T cells are thought to play a pivotal role in the mechanism of RRD during immunotherapy, potentially acting as mediators of the immune reaction to checkpoint inhibitors. Alotaibi et al. [20] demonstrated that blocking CD5 can enhance T cell-mediated anti-tumor immunity and slow tumor growth in mice, indicating CD5 may also play a key role in regulating immune activation. In the presented case, a pronounced infiltration of CD5^+^ T-lymphocytes was observed. Additionally, Korenfeld et al. [21] identified a subset of human skin dendritic cells expressing CD5, which are associated with inflammatory skin diseases. This suggests that CD5^+^ cells in the skin may include both T cells and dendritic cells contributing to the local immune environment and exacerbating the recall phenomenon in previously irradiated tissue. Although few cases have been reported on the pathological changes within the dermis and epidermis, further studies may help clarify the underlying mechanisms of immune checkpoint inhibitor-induced RRD and improve our understanding of its pathophysiology.

Due to the rarity of RRD with ICIs, there are no established guidelines for managing these cases. The severity of symptoms should be considered, as the reports indicate that most patients will ultimately improve. Treatment mirroring that in irAEs in the absence of radiation recall seems prudent. Topical steroids for mild cases may be all that is required [12,14]. For more severe cases involving necrosis or desquamation of the skin, treatment may require both systemic and topical steroids [13,15]. The benefits versus risks of withholding ICI following RRD and whether a patient would then be rechallenged if the drug is held must be individualized. A few patients were able to continue using ICIs without interruption, including our patient [5]. Some patients resumed the drug after a period of interruption, with mixed outcomes [5,14]. Our patient received a short course of prednisolone with rapid improvement in dermatitis and was able to continue pembrolizumab without interruption, granted with a different dose and frequency modification.

The absence of disease progression in our patient despite the severe dermatologic reaction presents a unique perspective on the interaction between RRD and the underlying malignancy. Previous reports have variably associated RRD with both stable disease and periods of tumor activity. As such, further studies are needed to delineate whether the occurrence of RRD might correlate with the immunological control of metastatic disease, potentially serving as a biomarker for the effectiveness of ICI therapy. Understanding this relationship could guide more tailored treatment approaches and provide insights into the complex dynamics between radiotherapy-induced tissue changes and immune activation.

## 4. Conclusions

The present case illustrates the variability in the presentation of RRD from ICIs, in particular the potential to develop this reaction years after radiation exposure. Clinical features mimicked those seen in inflammatory breast cancer. On tissue biopsy, T-lymphocyte infiltration was predominant in the dermis of this current patient. The inflammatory process responded quickly to corticosteroid therapy. ICI treatment with pembrolizumab was continued on schedule, although subsequent dosing was given at 3-week intervals rather than 6 weeks, with a corresponding dose reduction from 400 mg to 200 mg per treatment. There has been no recurrence of RRD in this patient following steroid withdrawal.

Our case highlights the potential for RRD to occur without concurrent cancer progression, raising intriguing questions about the immunological underpinnings of this phenomenon. It is essential to further investigate how disease progresses in patients who experience RRD following ICI therapy, as this could greatly influence patient management and potentially offer a predictive value regarding treatment responses. Additional studies are necessary to elucidate the pathophysiology and identify risk factors associated with RRD in the context of ICI treatments. Although cases of RRD associated with ICIs have, to this point, been infrequently reported, the incidence is likely to increase given the broadening applications of immunotherapy in many tumor types. Prompt recognition of this entity and an appropriate measured response may both limit serious sequelae from the reaction and simultaneously allow for the continuation of beneficial therapy.

## Figures and Tables

**Figure 1 reports-07-00091-f001:**
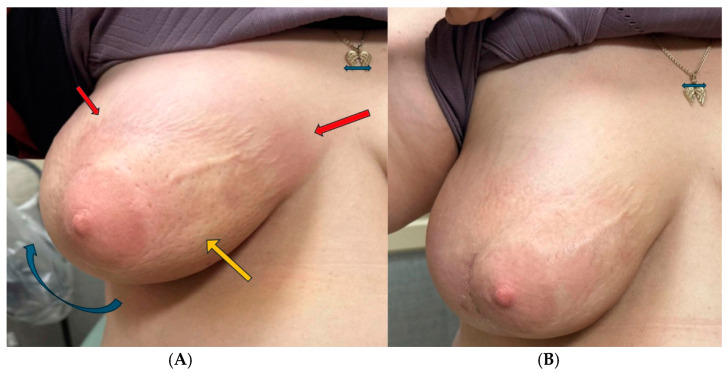
Right breast radiation recall dermatitis (RRD) (**A**). Recovery of RRD at eight weeks following systemic steroids (**B**). RRD with generalized induration and retraction of breast [blue arrow], peau d’orange texture [orange arrow], erythema, wrinkling, and discoloration [red arrow]. The green arrow over the pendant in both photos is equal in length and serves as a reference to illustrate that magnification in the photos are identical. Retraction of breast and peau d’orange resolved. Wrinkling and erythema reduced. Complete resolution of all changes over the subsequent four weeks.

**Figure 2 reports-07-00091-f002:**
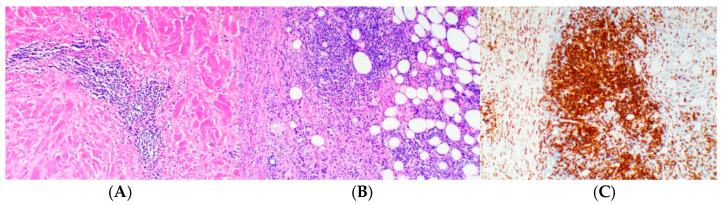
H&E low power view (100×) showing perivascular inflammation and marked dermal fibrosis (**A**). H&E intermediate power (200×) showing marked mixed inflammatory response in dermis (**B**). CD5^+^ stain (100×) showing predominant T-cell mediated response in dermis (Leica Biosystems, Buffalo Grove, IL, USA) (**C**).

## Data Availability

The original data presented in this study are available on reasonable request from the corresponding author. The data are not publicly available due to privacy.

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
