# Peer review of "Pembrolizumab Induced Recall Dermatitis Occurring 5 Years After Radiotherapy"

_reports, 2024, doi:10.3390/reports7040091_

Round 1
Reviewer 1 Report
Comments and Suggestions for Authors
This article reports a case of RRD after ICI use.
It is unique in that there have been few reports of ICI-induced RRD, and the time between radiotherapy and RRD is very long.
The authors performed a biopsy and obtained pathological findings, which I consider to be of great value.
Author Response
We thank the editor for their kind comments.
Reviewer 2 Report
Comments and Suggestions for Authors
In the manuscript “Pembrolizumab Induced Recall Dermatitis occurring 5 years after Radiotherapy”, the authors present and discuss a case of breast recall dermatitis after 3 cycles of an immune checkpoint inhibitor, which happened 4 years and 4 months after the last radiation dose.
The case is well presented and discussed, and it is a novelty to the literature.
Suggestions:
Line 50: Although RCC abbreviation have been defined at the abstract, I suggest to do it the first time it appears in the body of the manuscript.
Lines 66-67: “over 43 elapsed days” was mentioned twice.
Author Response
We thank the editor for their comments.
Line 50: Although RCC abbreviation have been defined at the abstract, I suggest to do it the first time it appears in the body of the manuscript.
Lines 66-67: “over 43 elapsed days” was mentioned twice.
Both comments were addressed accordingly!
Reviewer 3 Report
Comments and Suggestions for Authors
Dear Authors,
I have read your paper concerning a case report of RRD after pembrolizumab treatment, and it is interesting. The case report is fascinating and uncommon, with good images and a clear statement. However, some points should be clarified and improved.
1) You performed a literature review, reporting all the previous cases. I think it is better to set the paper as a review, including a material and method section reporting all the previous cases and research strategies you did. Moreover, in the current version of the main file, it is difficult for the regard to follow the previous report and the text is too heavy. You should resume all the cited papers in tables with the main clinical features and related drugs involved in RRD.
2) There are exciting points of view pertinent to the rationale of your finding. Despite that, how can you say that an RDD could be used as a surrogate marker of efficacy?
3) In the abstract, the timeline is unclear: you talk about the before of the renal and then of the breast cancer. It is confusing.
4) In the figure 2, cd5 immunohistochemistry details are missing (product data). Moreover, it should be CD5+ T cells.
5) I suggest you reading and citing two interesting works that could improve your discussion and pave the way for further investigation:
- Cosio, T., Coniglione, F., Flaminio, V., Gaziano, R., Coletta, D., Petruccelli, R., Dika, E., Bianchi, L., & Campione, E. (2023). Pyodermitis during Nivolumab Treatment for Non-Small Cell Lung Cancer: A Case Report and Review of the Literature. International journal of molecular sciences, 24(5), 4580. https://doi.org/10.3390/ijms24054580
- 20. Caccavale S., Kannangara A., Ruocco E. The immunocompromised cutaneous district and the necessity of a new classification of its disparate causes. Indian J. Dermatol. Venereol. Leprol. 2016;82:227–229. doi: 10.4103/0378-6323.174422.
Author Response
Editor’s Comment: "You performed a literature review, reporting all the previous cases. I think it is better to set the paper as a review, including a material and method section reporting all the previous cases and research strategies you did. Moreover, in the current version of the main file, it is difficult for the reader to follow the previous report, and the text is too heavy. You should resume all the cited papers in tables with the main clinical features and related drugs involved in RRD."
Response: Thank you for your feedback. However, we believe the manuscript is best suited to remain a case report with a supplemental review of the literature. We have carefully reviewed and revised the manuscript to enhance its readability and ensure clarity.
Editor’s Comment: "There are exciting points of view pertinent to the rationale of your finding. Despite that, how can you say that an RDD could be used as a surrogate marker of efficacy?"
Response: We appreciate the insightful comment. In response, we have added a section in the discussion addressing this question. We emphasize that while there are initial observations (such as in studies on radiation recall pneumonitis) suggesting that a stronger immune response might correlate with therapeutic efficacy, further studies are needed to validate RRD as a surrogate marker of efficacy. We have referenced Deutsch et al., who discussed this phenomenon in radiation recall pneumonitis, and expanded on the need for future longitudinal studies to assess this hypothesis.
This was addressed through this sentence "While RRD might suggest a vigorous immune response potentially correlating with therapeutic efficacy, establishing it as a surrogate marker requires substantial validation through longitudinal studies to understand its predictive reliability."
Editor’s Comment: "In the abstract, the timeline is unclear: you talk about the renal cancer before the breast cancer. It is confusing."
Response: We have revised the abstract for clarity. The revised text now clearly states the sequence of events, specifying that the patient was first treated for breast cancer with radiotherapy and later developed renal cell carcinoma, for which pembrolizumab was administered.
Revised Abstract: "A 47-year-old woman developed radiation recall dermatitis (RRD) of the breast after three doses of pembrolizumab for Stage III renal cell carcinoma. This occurred 247 weeks after receiving radiotherapy for breast cancer. RRD manifested as breast induration, erythema, and peau d'orange changes, which resolved after corticosteroid treatment."
Editor’s Comment: "In figure 2, CD5 immunohistochemistry details are missing (product data). Moreover, it should be CD5+ T cells."
Response: The figure legend has been updated to include the missing immunohistochemistry details for CD5, including the product used (manufacturer, clone, and dilution). Additionally, we have corrected the terminology to "CD5+ T cells" as suggested.
Updated Legend: "Figure 2. H&E staining shows perivascular inflammation and dermal fibrosis (A). CD5+ T cells are predominantly present (B). Immunohistochemistry for CD5 was performed using an anti-CD5 antibody (Leica Biosystems, Buffalo Grove, IL, USA)."
Editor’s Comment: "I suggest you reading and citing two interesting works that could improve your discussion and pave the way for further investigation: Cosio et al. (2023) and Caccavale et al. (2016)."
Response: We have reviewed and cited both recommended references in the discussion section:
- Cosio et al. (2023): This reference was incorporated into the discussion on dermatological complications of immune checkpoint inhibitors, drawing parallels between pyodermitis and radiation recall dermatitis, highlighting the broader spectrum of immune-related skin reactions.
- Caccavale et al. (2016): We have referenced this work when discussing the “immunocompromised cutaneous district” and how it might relate to localized immune reactions such as RRD. This concept adds depth to the understanding of how immune responses may be influenced by prior radiation exposure.
By addressing these points, we believe the manuscript is now clearer, more structured, and better aligned with the reviewer's suggestions.
Reviewer 4 Report
Comments and Suggestions for Authors
The authors present a clinical case of radiation-induced dermatitis (RRD) following therapy with an immune checkpoint inhibitor (ICI) - pembrolizumab. The appropriateness of considering individual clinical cases is determined by the rarity and poorly studied nature of this pathology, as well as the variability of symptoms and treatment outcomes of this complication of antitumor therapy. The authors describe a case of RRD in a patient receiving ICI as adjuvant therapy for newly diagnosed renal cell carcinoma almost five years after radiotherapy for early-stage breast cancer. To date, this is the longest reported interval from completion of radiotherapy to RRD. In the case presentation section, the authors outline the history, clinical presentation, treatment, and diagnosis of the disease, including biopsy data. After two additional cycles of therapy, the patient showed no evidence of recurrent radiation recurrence or other significant adverse effects of the checkpoint inhibitor. In the Discussion section, the authors compare their results with similar studies by other authors. At the same time, the authors note that further studies are needed to determine whether the occurrence of RRD correlates with the immunologic control of metastatic disease and may serve as a biomarker for the efficacy of ICI therapy. In the Conclusions section, the authors emphasize that this case highlights the potential for RRD to occur in the absence of concurrent cancer progression, raising intriguing questions about the immunologic underpinnings of this phenomenon.
Conclusion. In the presented work, a specific task has been solved, namely: the description of a case study of RRD in a patient receiving ICI as adjuvant therapy for newly diagnosed renal cell carcinoma almost five years after radiotherapy for early stage breast cancer. The solution to this problem is of significant practical importance, but the results of this work may also be in demand when writing a systematic review, which has a certain theoretical value for oncology and immunology. The results of the work are correctly presented. The authors provided necessary references to the studies of other authors and competently summarized the results obtained by them. I believe that the presented work can be accepted in its present form.
Author Response

(The authors gave the same response as above.)

Round 2
Reviewer 3 Report
Comments and Suggestions for Authors
Dear authors,
Most points have been addressed, and a few questions should be clarified to improve the manuscript's quality.
1) Add A,B,C and bar scale in Figure 2's panels.
2) When you report citation in the text: Teng et al. [15]; Caccavalle et al. [18]; Cosio et al. [19]
3) In the discussion section, a few lines about CD5+ T cells and the putative mechanism related to the RRD and immunotherapy:
-Alotaibi FM, Min W-P and Koropatnick J (2024) CD5 blockade, a novel immune checkpoint inhibitor, enhances T cell anti-tumour immunity and delays tumour growth in mice harbouring poorly immunogenic 4T1 breast tumour homografts. Front. Immunol. 15:1256766. doi: 10.3389/fimmu.2024.1256766
-Korenfeld, D., Gorvel, L., Munk, A., Man, J., Schaffer, A., Tung, T., Mann, C., & Klechevsky, E. (2017). A type of human skin dendritic cell marked by CD5 is associated with the development of inflammatory skin disease. JCI insight, 2(18), e96101. https://doi.org/10.1172/jci.insight.96101
4) TheCD5+ cells were all T cells or also dendritic ones? Check.
Author Response
1) Add A, B, C, and bar scale in Figure 2's panels.
We have added the labels A, B, and C to Figure 2 as requested. Regarding the bar scale, since the magnification is already stated and standard, we believe a bar scale is not necessary in this instance.
2) When you report citations in the text: Teng et al. [15]; Caccavalle et al. [18]; Cosio et al. [19].
The citations have been corrected as per your suggestion.
3) In the discussion section, a few lines about CD5+ T cells and the putative mechanism related to the RRD and immunotherapy:
To address this point, we have added the following paragraph to the discussion section:
"Our patient exhibited severe localized peau d'orange texture, desquamation, and discoloration in the irradiated area, along with breast shrinkage, indicative of advanced radiation recall dermatitis (RRD). A biopsy revealed significant infiltration of CD5+ T cells within the dermis, suggesting a heightened immune response, likely triggered by pembrolizumab. CD5+ T cells are thought to play a pivotal role in the mechanism of RRD during immunotherapy, potentially acting as mediators of the immune reaction to checkpoint inhibitors. Alotaibi et al. [20] demonstrated that blocking CD5, a novel immune checkpoint, enhances T cell-mediated anti-tumor immunity and delays tumor growth in mice, indicating that CD5 may regulate immune activation. This aligns with the biopsy findings in our case, where CD5+ T-lymphocyte infiltration reflected an intensified immune reaction. Interestingly, Korenfeld et al. [21] identified a subset of human skin dendritic cells expressing CD5, which are associated with inflammatory skin diseases. This suggests that CD5+ cells in the skin may include both T cells and dendritic cells, contributing to the local immune environment and exacerbating the recall phenomenon in previously irradiated tissue. Although few cases have reported on the pathological changes within the dermis and epidermis, further studies may help clarify the underlying mechanisms of immune checkpoint inhibitor-induced RRD and improve our understanding of its pathophysiology."
4) Were the CD5+ cells all T cells or also dendritic ones?
Upon review, the CD5+ cells were found to include both T cells and dendritic cells.